# International multicenter study comparing COVID-19 in patients with cancer to patients without cancer: Impact of risk factors and treatment modalities on survivorship

Issam I Raad[1†], Ray Hachem[1†], Nigo Masayuki[2], Tarcila Datoguia[3], Hiba Dagher[1], Ying Jiang[1], Vivek Subbiah[4,5], Bilal Siddiqui[6], Arnaud Bayle[7], Robert Somer[8], Ana Fernández Cruz[9], Edward Gorak[10], Arvinder Bhinder[11], Nobuyoshi Mori[12], Nelson Hamerschlak[3], Samuel Shelanski[13], Tomislav Dragovich[14], Yee Elise Vong Kiat[15], Suha Fakhreddine[16], Abi Hanna Pierre[16], Roy F Chemaly[1], Victor Mulanovich[1], Javier Adachi[1], Jovan Borjan[1], Fareed Khawaja[1], Bruno Granwehr[1], Teny John[1], Eduardo Yepez Yepez[1], Harrys A Torres[1], Natraj Reddy Ammakkanavar[5], Marcel Yibirin[1], Cielito C Reyes-Gibby[17], Mala Pande[18], Noman Ali[19], Raniv Dawey Rojo[20], Shahnoor M Ali[1], Rita E Deeba[1], Patrick Chaftari[17], Takahiro Matsuo[12], Kazuhiro Ishikawa[12], Ryo Hasegawa[12], Ramón Aguado-Noya[21], Alvaro Garcia García[22], Cristina Traseira Puchol[21], Dong Gun Lee[23], Monica Slavin[24], Benjamin Teh[24], Cesar A Arias[2], Data-Driven Determinants for COVID-19 Oncology Discovery Effort (D3CODE) Team[25], Dimitrios P Kontoyiannis[1], Alexandre E Malek[1], Anne-Marie Chaftari[1*†]

[1]Department of Infectious Diseases, Infection Control and Employee Health, The University of Texas MD Anderson Cancer Center, Houston, United States; [2]Division of Infectious Diseases, McGovern Medical School, The University of Texas Health Science Center at Houston, Houston, United States; [3]Médica Hematologista Hospital Israelita Albert Einstein, São Paulo, Brazil; [4]MD Anderson Cancer Network, UT MD Anderson Cancer Center, Houston, United States; [5]Department of Investigational Cancer Therapeutics, The University of Texas MD Anderson Cancer Center, Houston, United States; [6]Department of Hematology Oncology, Community Health Network, Indianapolis, United States; [7]Department of Medical Oncology, Gustave Roussy, Université Paris-Saclay, Villejuif, France; [8]Cooper Medical School of Rowan University, Cooper University Health Care, Camden, United States; [9]Unidad de Enfermedades Infecciosas, Servicio de Medicina Interna, Hospital Universitario Puerta de Hierro, Madrid, Spain; [10]Department of Hematology Oncology, Baptist Health, Jacksonville, United States; [11]Department of Hematology/Oncology, Ohio Health Marion, Marion, United States; [12]Department of Infectious Diseases, St. Luke's International Hospital, Tokyo, Japan; [13]Banner MD Anderson Cancer Center – North Colorado, Greely, United States; [14]Division of Cancer Medicine, Banner MD Anderson Cancer Center, Gilbert, United States; [15]Department of Medical Oncology, Tan Tock Seng Hospital, Singapore, Singapore; [16]Department of Infectious Diseases, Rafik Hariri University Hospital, Beirut, Lebanon; [17]Department of Emergency Medicine, The University of Texas MD Anderson Cancer Center, Houston, United States; [18]Department of Gastroenterology, The University of Texas MD Anderson Cancer Center, Houston,

*For correspondence:
achaftari@mdanderson.org

†These authors contributed equally to this work

United States; [19]Department of Hospital Medicine, The University of Texas MD Anderson Cancer Center, Houston, United States; [20]Department of Breast Surgical Oncology, The University of Texas MD Anderson Cancer Center, Houston, United States; [21]Oncology Department, Hospital Universitario Puerta de Hierro-Majadahonda, Madrid, Spain; [22]Hematology Department, Hospital Universitario Puerta de Hierro-Majadahonda, Madrid, Spain; [23]Division of Infectious Diseases, Department of Internal Medicine, Vaccine Bio Research Institute, The Catholic University of Korea, Seoul, Republic of Korea; [24]Department of Infectious Diseases and National Centre for Infections in Cancer, Peter MacCallum Cancer Centre, Melbourne, Australia; [25]Data-Driven Determinants for COVID-19 Oncology Discovery Effort (D3CODE) Team at The University of Texas MD Anderson Cancer Center, Houston, United States

## Abstract

**Background:** In this international multicenter study, we aimed to determine the independent risk factors associated with increased 30 day mortality and the impact of cancer and novel treatment modalities in a large group of patients with and without cancer with COVID-19 from multiple countries.

**Methods:** We retrospectively collected de-identified data on a cohort of patients with and without cancer diagnosed with COVID-19 between January and November 2020 from 16 international centers.

**Results:** We analyzed 3966 COVID-19 confirmed patients, 1115 with cancer and 2851 without cancer patients. Patients with cancer were more likely to be pancytopenic and have a smoking history, pulmonary disorders, hypertension, diabetes mellitus, and corticosteroid use in the preceding 2 wk (p≤0.01). In addition, they were more likely to present with higher inflammatory biomarkers (D-dimer, ferritin, and procalcitonin) but were less likely to present with clinical symptoms (p≤0.01). By country-adjusted multivariable logistic regression analyses, cancer was not found to be an independent risk factor for 30 day mortality (p=0.18), whereas lymphopenia was independently associated with increased mortality in all patients and in patients with cancer. Older age (≥65y) was the strongest predictor of 30 day mortality in all patients (OR = 4.47, p<0.0001). Remdesivir was the only therapeutic agent independently associated with decreased 30 day mortality (OR = 0.64, p=0.036). Among patients on low-flow oxygen at admission, patients who received remdesivir had a lower 30 day mortality rate than those who did not (5.9 vs 17.6%; p=0.03).

**Conclusions:** Increased 30 day all-cause mortality from COVID-19 was not independently associated with cancer but was independently associated with lymphopenia often observed in hematolgic malignancy. Remdesivir, particularly in patients with cancer receiving low-flow oxygen, can reduce 30 day all-cause mortality.

**Funding:** National Cancer Institute and National Institutes of Health.

## Editor's evaluation

This study addresses the important subject of correlates of COVID-19 mortality in patients with cancer. It is a large multinational cohort of patients. The authors provide solid evidence based on careful multivariate analysis for the finding that cancer is not an independent risk factor for COVID-19 death.

## Introduction

The COVID-19 pandemic has challenged the health care system worldwide and has spread to more than 200 countries, causing hundreds of millions confirmed cases and several million deaths (*Johns Hopkins Coronavirus Resource Center, 2021*).

Data from multiple studies have shown consistently that older age and comorbidities such as cardiovascular disease, diabetes mellitus (DM), hypertension, and chronic obstructive pulmonary disease (COPD) have been associated with severe illness and increased mortality (*Zhou et al., 2020*).

Several studies on COVID-19 mortality suggested that patients with cancer had poor outcomes (*Kuderer et al., 2020*; *Rivera et al., 2020*; *He et al., 2020*; *Albiges et al., 2020*; *Robilotti et al., 2020*; *Lee et al., 2020*; *Mehta et al., 2020*; *Tian et al., 2020*; *Lunski et al., 2021*; *Sun et al., 2020*). Many of these studies included only patients with cancer and did not have a comparator group of patients without cancer (*Kuderer et al., 2020*; *Rivera et al., 2020*; *He et al., 2020*; *Albiges et al., 2020*; *Robilotti et al., 2020*). Other studies compared COVID-19 mortality between patients with and without cancer and found that patients with cancer had worse outcomes (*Mehta et al., 2020*; *Tian et al., 2020*; *Lunski et al., 2021*; *Sun et al., 2020*; *Rüthrich et al., 2021*). However, all of these comparative studies included a relatively small number of patients with cancer and were restricted to a particular country, which limits their generalizability to patients with cancer worldwide.

Given that many of the therapeutic studies on COVID-19 were conducted in patients without cancer and given the poor outcomes of COVID-19 reported in patients with cancer (*Mehta et al., 2020*; *Tian et al., 2020*; *Lunski et al., 2021*; *Sun et al., 2020*; *Rüthrich et al., 2021*; *Beigel et al., 2020*; *Salazar et al., 2021*; *Libster et al., 2021*; *Janiaud et al., 2021*; *Horby et al., 2021*), we aimed to conduct a large multicenter study to compare the impact of these various treatments on the outcome in patients with cancer vs patients without cancer.

Therefore, given the global spread of the COVID-19 pandemic and the worldwide prevalence of cancer, we undertook this international initiative that included 16 centers from five continents to study and compare the clinical course, risk factors, and treatment modalities impact on outcomes of COVID-19 in patients with cancer vs patients without cancer on a worldwide basis.

## Methods

### Study design and participants

This was a retrospective international multicenter study that included all patients diagnosed with COVID-19 by RT-PCR for severe acute respiratory syndrome coronavirus 2 (SARS-CoV-2) at the site or an outside facility between January 4, 2020 and November 15, 2020.

The study involved 16 centers from 9 countries, 8 centers in the United States and 1 each in Australia, Brazil, France, Japan, Lebanon, Singapore, South Korea, and Spain. Patients were divided into two groups: patients without cancer and those with cancer diagnosed or treated within a year before the diagnosis of COVID-19.

### Multicenter collaboration and data collection

The University of Texas MD Anderson Cancer Center was the coordinating center that designed the study, built the electronic case report form, and collected de-identified patient information from all participating centers using the secure Research Electronic Data Capture platform. We reviewed each patient's electronic hospital record and collected all needed data. This study was approved by the institutional review board at MD Anderson Cancer Center (Protocol# 2020–0437) and the institutional review boards of the collaborating centers. A patient waiver of informed consent was obtained.

The follow-up period was defined as 30 days after a diagnosis of COVID-19.

Neutropenia was defined as an absolute neutrophil count <500 cells/mL. Lymphopenia was defined as an absolute lymphocyte count (ALC) <500 cells/mL.

### Treatment modalities

For each patient, data were collected on COVID-19 treatments, including antimicrobial therapy and potential antiviral therapy. We evaluated patient's oxygen requirement: low flow was defined as oxygen supplementation of ≤6 l/min through a nasal cannula or facemask and high flow included all other modalities of oxygen supplementation, including mechanical ventilation.

### Outcomes measures

The primary outcome of interest was 30 day mortality. Any death that occurred within 30 days after COVID-19 diagnosis was considered to be COVID-19-related, irrespective of other comorbidities that

could have contributed to death. The secondary outcomes included mechanical ventilation, progression to lower respiratory tract infection, co-infection, and hospital readmissions within 30 days after COVID-19 diagnosis.

### Statistical analysis

Patient characteristics and outcomes were compared between COVID-19 patients with and without cancer. Categorical variables were compared using $\chi^2$ or Fisher's exact test, and continuous variables were compared using Wilcoxon rank sum test. Logistic regression model was used to identify the factors that were independently associated with 30 day mortality, and the following factors were included in the analysis in addition to cancer status: patients' demographic (including country) and clinical characteristics, medical history, and laboratory findings at diagnosis and treatment. First, univariable analysis of each factor was performed, but factors were not considered if more than 40% of the data were missing. Next all the factors with p-values≤0.15 on their univariable analyses were included in a full logistic regression model and then the full model was reduced to the final model by backward elimination procedure so that all the factors remaining in the final model had p-values≤0.05 except cancer and country. Cancer was kept in the final model despite its p-value in order to evaluate its independent impact on mortality. The final model was country-adjusted to account for the treatment differences among the countries. For the above complete cases analysis, only patients with no missing data in any of the variables retained for the final regression model were included in the final model analysis. To investigate the impact of missing data on our primary data analysis, we performed a sensitivity analysis. We first estimated missing values using multiple imputation technique and then performed a similar multivariable analysis based on the imputed datasets. Lastly, the analysis results were compared between the complete-case and multiple imputation analyses. Logistic regression model was also used to identify the independent predictors of mortality among patients with and without cancer, respectively, and similar sensitivity analysis was performed for each group as well. In cancer patients, the mortality rates of four different cancers were estimated and compared: hematological malignancy, lung cancer, and non-lung solid tumors with and without metastasis. $\chi^2$ or Fisher's exact test was used for comparisons. If a significant result (p<0.05) was detected, pairwise comparisons were performed with α levels adjusted using Holm's sequential Bonferroni adjustment to control type I error. In addition, the associations between mortality and certain therapeutic agents including remdesivir, steroids, and convalescent plasma, as well as the impact of their treatment timing on mortality, were also evaluated using $\chi^2$ or Fisher's exact test. All statistical tests were two-sided with a significance level of 0.05, except the pairwise comparisons with the α adjustment. Statistical analyses were performed using SAS version 9.4 (SAS Institute Inc, Cary, NC, USA).

## Results

A total of 4015patients diagnosed with COVID-19 by PCR were included in the study. After excluding 18 patients who had missing demographics and 31 patients younger than 18y, we evaluated 3966 COVID-19 patients: 2851 without cancer and 1115 with cancer (see *Figure 1* for a consort flow diagram of patient attrition in analyses).

### Demographics and clinical characteristics of patients with and without cancer

Patient characteristics are presented in *Table 1*. Compared to patients without cancer, patients with cancer were older (median age, 61 vs 50 y; p<0.0001); more likely to have a smoking history (38 vs 17%; p<0.0001), pulmonary disorder (27 vs 21%; p<0.001), hypertension (49 vs 36%, p<0.0001), DM (27 vs 23%, p=0.01), or corticosteroid use in the 2 wk preceding COVID-19 diagnosis (17 vs 4%, p<0.0001); and less likely to present with clinical symptoms, including cough (46 vs 65%; p<0.0001), fever (45 vs 66%; p<0.0001), and shortness of breath (35 vs 48%; p<0.0001).

Compared to patients without cancer, patients with cancer were more likely to present with neutropenia (7 vs 0.1%; p<0.0001), lymphocytopenia (49 vs 32%; p<0.0001), thrombocytopenia (39 vs 13%; p<0.0001), and anemia (47 vs 29%; p<0.0001) and had higher median levels of inflammatory biomarkers, including D-dimer (1.95 vs 1.51 μg/ml, p=0.013), ferritin (1015 vs 823 ng/ml, p<0.0001), and procalcitonin (0.25 vs 0.21 ng/ml, p=0.008). On imaging studies (CT), patients with cancer were

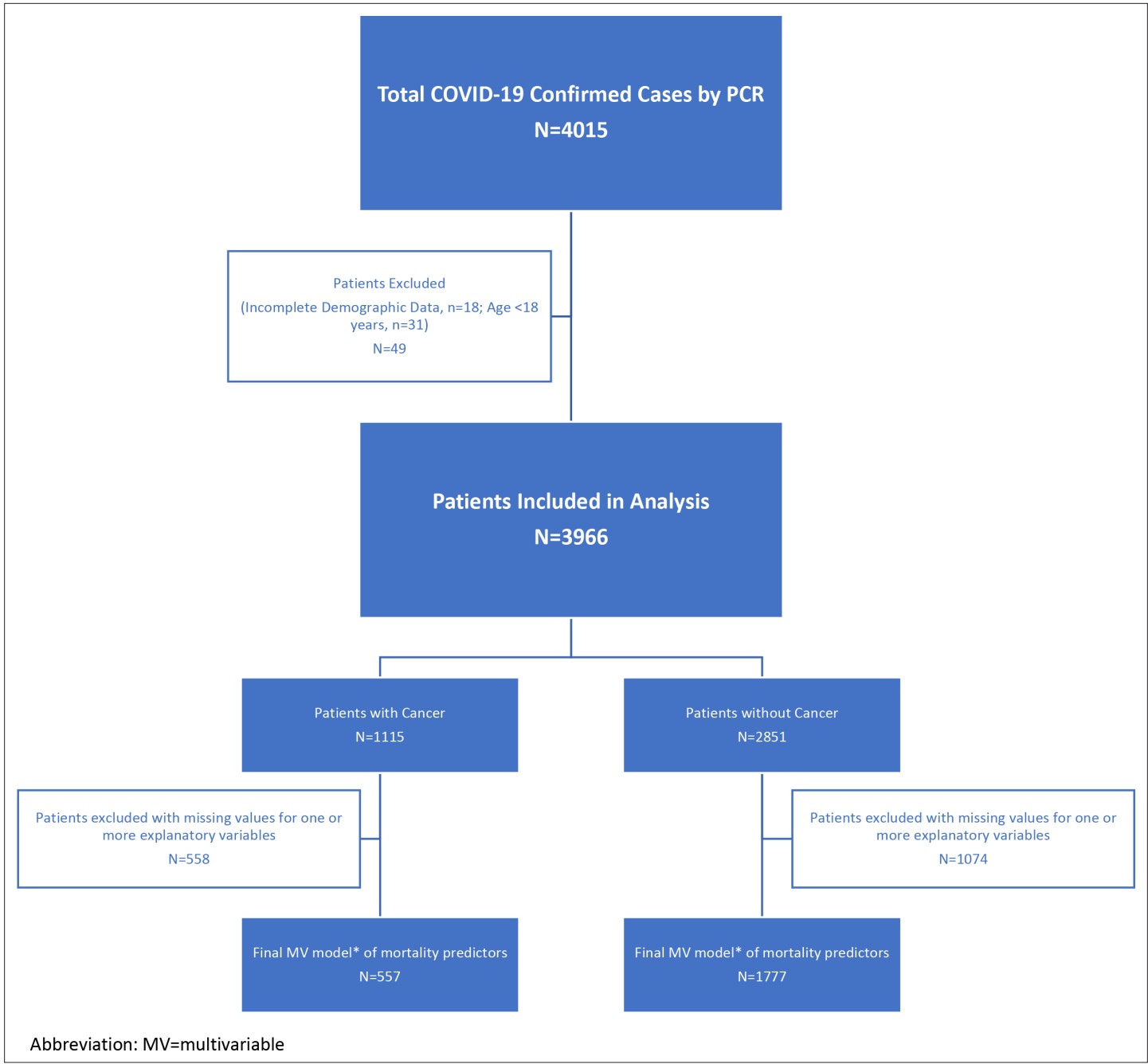

**Figure 1.** Consort diagram of patient attrition.

less likely than patients without cancer to be present with ground-glass opacities (68 vs 78%; p<0.0001) or peripheral distribution of the infiltrates (17 vs 41%; p<0.0001).

## Treatment and outcomes of patients with and without cancer

For most treatments for COVID-19 infection, patients without cancer were more likely to receive them than patients with cancer, including hydroxychloroquine (23 vs 18%, p<0.001), azithromycin (47 vs 18%, p<0.0001), remdesivir (12 vs 9%, p=0.004), convalescent plasma (12 vs 6%, <0.0001), and steroids (43 vs 17%, p<0.0001). However, patients with cancer were more likely to receive tocilizumab (6 vs 4%, p=0.002). In addition, the treatments patients received for COVID-19 infection significantly differed among the countries (data not shown). The rate of COVID-related hospital admission was higher in the patients without cancer (62 vs 45%; p<0.0001). Co-infections occurred more frequently

**Table 1.** Comparing COVID-19 patients with and without cancer.

| Characteristic | Without cancer (n=2851) N (%) | With cancer (n=1115) N (%) | p-value |
|---|---|---|---|
| Demographic and baseline clinical characteristics | | | |
| Age (years), median (range) | 50 (18–100) | 61 (18–100) | <0.0001 |
| Sex, male | 1335 (47) | 506 (45) | 0.41 |
| Race/ethnicity | | | <0.0001 |
| White | 720/2674 (27) | 534/916 (58) | |
| Black | 517/2674 (19) | 155/916 (17) | |
| Hispanic | 547/2674 (20) | 178/916 (19) | |
| Asian | 275/2674 (10) | 33/916 (4) | |
| Middle Eastern | 63/2674 (2) | 3/916 (0.3) | |
| Other | 552/2674 (21) | 13/916 (1) | |
| Prior pulmonary disorders | 431/2064 (21) | 275/1017 (27) | <0.001 |
| COPD/bronchiolitis obliterans | 175/2054 (9) | 75 (7) | 0.07 |
| Asthma | 178 (6) | 106 (10) | <0.001 |
| Obstructive sleep apnea | 98/2054 (5) | 89/948 (9) | <0.0001 |
| History of heart failure | 240/2036 (12) | 85/1098 (8) | <0.001 |
| History of ischemic heart disease | 226/2830 (8) | 94/1101 (9) | 0.57 |
| History of hypertension | 1020/2837 (36) | 546/1110 (49) | <0.0001 |
| History of diabetes mellitus | 659/2837 (23) | 299/1104 (27) | 0.01 |
| Current or previous smoker | 348/2055 (17) | 409/1066 (38) | <0.0001 |
| Corticosteroid treatment within 2wk prior to COVID-19 diagnosis | 40/1041 (4) | 189/1097 (17) | <0.0001 |
| Presenting symptoms | 981/1061 (92) | 813/1102 (74) | <0.0001 |
| Cough | 685/1061 (65) | 505/1102 (46) | <0.0001 |
| Fever | 698/1061 (66) | 498/1102 (45) | <0.0001 |
| Shortness of breath | 508/1061 (48) | 387/1102 (35) | <0.0001 |
| Chest pain | 100/1061 (9) | 66/1102 (6) | 0.003 |
| Headache | 133/1061 (13) | 80/1102 (7) | <0.0001 |
| Gastrointestinal symptoms | 148/1061 (14) | 104/1102 (9) | 0.001 |
| Loss of smell | 84/1061 (8) | 58/1102 (5) | 0.013 |
| Loss of taste | 78/1061 (7) | 55/1102 (5) | 0.022 |
| ICU admission | 225/1846 (12) | 141/1100 (13) | 0.62 |
| Abnormal laboratory values | | | |
| ANC <0.5K/µl | 2/1434 (0.1) | 30/419 (7) | <0.0001 |
| ALC <0.5K/µl | 487/1537 (32) | 225/463 (49) | <0.0001 |
| Platelet count<100K/µl | 187/1446 (13) | 151/387 (39) | <0.0001 |
| Hemoglobin<10g/dL | 437/1533 (29) | 283/601 (47) | <0.0001 |
| D-dimer, median (range), µg/ml | 1.51 (0.04–735.0) | 1.95 (0.25–93.24) | 0.013 |

*Table 1 continued on next page*

*Table 1 continued*

| Characteristic | Without cancer (n=2851) N (%) | With cancer (n=1115) N (%) | p-value |
|---|---|---|---|
| Ferritin, median (range), ng/ml | 823 (1.10–89672) | 1015 (1.5–100001) | <0.0001 |
| Procalcitonin, median (range), ng/ml | 0.21 (0.009–163.5) | 0.25 (0–101.8) | 0.008 |
| IL-6, median (range), pg/ml | 61 (0–5001) | 44 (0–7525) | 0.13 |
| Imaging findings | | | |
| New infiltrates | 151/622 (24) | 138/424 (33) | 0.003 |
| Ground-glass opacities | 487/622 (78) | 287/425 (68) | <0.0001 |
| Peripheral distribution of infiltrates | 253/622 (41) | 60/347 (17) | <0.0001 |
| Treatment | | | |
| Hydroxychloroquine | 475/2054 (23) | 196/1114 (18) | <0.001 |
| Azithromycin | 972/2054 (47) | 201/1114 (18) | <0.0001 |
| Remdesivir | 338 (12) | 97 (9) | 0.004 |
| Tocilizumab | 75/2054 (4) | 67 (6) | 0.002 |
| Convalescent plasma | 253/2054 (12) | 61/948 (6) | <0.0001 |
| Steroids | 879/2054 (43) | 192/1114 (17) | <0.0001 |
| Others* | 166/2054 (8) | 124/953 (13) | <0.0001 |
| Outcomes | | | |
| Co-infection after COVID-19 diagnosis | 158/1994 (8) | 116/1066 (11) | 0.006 |
| Multi-organ failure | 130/1052 (12) | 135/1096 (12) | 0.98 |
| Thrombotic complication | 56/1048 (5) | 48/1072 (5) | 0.36 |
| Discharged on supplemental oxygen among hospitalized patients | 90/776 (12) | 75/438 (17) | 0.007 |
| Hospital re-admission within 30 days of COVID-19 diagnosis | | | <0.0001 |
| No | 667/806 (83) | 318/481 (66) | |
| Yes | 60/806 (7) | 63/481 (13) | |
| Stayed in hospital (throughout 30 days) | 79/806 (10) | 100/481 (21) | |
| Death within 30 days of COVID-19 diagnosis | 226 (8) | 122 (11) | 0.003 |

Note:*Other treatment included chloroquine, favipiravir, lopinavir-ritonavir, anakina, baricitinib, type 1 interferons, and immunoglobulin.

Note: If a variable had missing data then the number of patients evaluable for this variable is added as denominator in its analysis result.

COPD = Chronic obstructive pulmonary disease. ANC = Absolute neutrophil count. ALC = Absolute lymphocyte count. IL-1=interleukin 1. IL-6=Interleukin 6. NA = Not applicable.

in patients with cancer (11 vs 8%; p=0.006). Among hospitalized patients, patients with cancer were more likely than patients without cancer to be discharged on supplemental oxygen (17 vs 12%; p=0.007). Likewise, the rate of hospital readmission within 30 days was higher in patients with cancer (13 vs 7%; p<0.0001). Furthermore, the mortality rate within 30 days was also significantly higher in patients with cancer by univariable analyses (11 vs 8%; p=0.003; *Table 1*).

**Table 2.** Country-adjusted multivariars of 30 day mortality among all patients.

| Independent predictor | Complete Case (CC) (N=2349) | | | Multiple imputation (MI) (N=3966) | | |
|---|---|---|---|---|---|---|
| | aOR | 95%CI | p-value | aOR | 95%CI | p-value |
| Age ≥65y | 4.47 | (3.27, 6.11) | <0.0001 | 4.73 | (3.54, 6.32) | <0.0001 |
| Prior COPD/bronchiolitis obliterans | 1.95 | (1.33, 2.85) | <0.001 | 1.73 | (1.21, 2.48) | 0.003 |
| History of heart failure | 1.61 | (1.13, 2.28) | 0.008 | 1.64 | (1.17, 2.29) | 0.004 |
| History of hypertension | 1.44 | (1.03, 2.01) | 0.036 | 1.52 | (1.11, 2.07) | 0.008 |
| Cancer | 1.30 | (0.89, 1.90) | 0.18 | 1.23 | (0.87, 1.75) | 0.24 |
| Hypoxia at diagnosis | 4.58 | (2.92, 7.19) | <0.0001 | 5.74 | (3.91, 8.45) | <0.0001 |
| Mechanical ventilation/intubation at diagnosis | 2.20 | (1.23, 3.93) | 0.008 | 2.23 | (1.30, 3.84) | 0.004 |
| ALC at diagnosis <0.5K/μl | 1.86 | (1.30, 2.64) | <0.001 | 1.79 | (1.27, 2.51) | <0.001 |
| Creatinine at diagnosis >1.5mg/dl | 1.68 | (1.21, 2.31) | 0.002 | 1.70 | (1.22, 2.38) | 0.002 |
| Hemoglobin at diagnosis <10g/dl | 1.54 | (1.06, 2.25) | 0.024 | 1.67 | (1.18, 2.38) | 0.004 |
| Coinfection after diagnosis | 1.83 | (1.25, 2.68) | 0.002 | 1.79 | (1.25, 2.56) | 0.001 |
| Remdesivir treatment | 0.64 | (0.42, 0.97) | 0.036 | | | |

The model was adjusted for country, tocilizumb treatment, and convalescent plasma treatment.

The significant difference between the models by CC analysis and by MI analysis was shown in the gray area - remdesivir treatment was an independent predictor of 30 day mortality in the multivariable model by CC analysis but not in the model by MI analysis.

COPD=Chronic obstructive pulmonary disease; ALC=Absolute lymphocyte count.

## Risk factors for death within 30 days after COVID-19 diagnosis

The independent predictors of 30 day mortality among all patients identified by the multivariable analysis are shown in *Table 2*. The multivariable complete-case analysis also showed that cancer was not an independent risk factor for 30 day mortality (p=0.18; *Table 2*). Older age (≥65y) was the strongest predictor of 30 day mortality in all patients (OR = 4.47; 95%CI=3.27–6.11; p<0.0001; *Table 2*) and in both patients with (OR = 6.64, p<0.0001) and without cancer (OR = 4.91, p<0.0001; *Table 3*).

Other independent risk factors for 30 day mortality in all patients included hypoxia at diagnosis (OR = 4.58; p<0.0001), need for mechanical ventilation (OR = 2.20; p=0.008), and presence of co-infection (OR = 1.83; p=0.002; *Table 2*).

In patients with cancer, lower respiratory tract infection manifested by the presence of pulmonary infiltrates either at diagnosis or during the course of infection was a strong independent predictor of 30 day mortality (OR = 3.70; 95% CI = 1.94–7.08; p<0.0001; *Table 3*).

Among patients with cancer, the 30 day mortality rate was significantly higher in patients with lung cancer (22%) than in patients with non-lung cancer solid tumors (6%, p<0.0001), including those with lung metastases (7%, p<0.001; *Table 4*). Patients with hematological malignancies had a significantly higher 30 day mortality than patients with non-lung cancer solid tumors (13 vs 6%, p<0.001) but tended to have a lower mortality rate than patients with lung cancer (13 vs 22%, p=0.07; *Table 4*). However, we did not find a significant difference in mortality between hematological malignancies and solid tumors by multivariable analysis (p=0.30).

By multivariable analysis, remdesivir was the only therapeutic agent independently associated with decreased 30 day all-cause mortality in all patients (OR = 0.64; 95% CI = 0.42–0.97; p=0.036; *Table 2*), and in patients with cancer (OR = 0.44; 95% CI = 0.20–0.96; p=0.04) as well (*Table 3*). However, in patients without cancer, remdesivir was not among the factors independently associated with mortality by multivariable analysis (*Table 3*).

Among patients on low-flow oxygen at admission, the mortality rate was lower among those who received remdesivir than among those who did not (5.9 [3 of 51] vs 17.6% [68 of 387]; p=0.03). However, among patients on high-flow oxygen at admission, there was no difference in the mortality

**Table 3.** Country-adjusted multivariable logistic regression analysis of independent predictors of 30 day mortality among different patients.

**A) Patients with cancer**

| Independent predictor | Complete case (CC) (N=557) | | | Multiple imputation (MI) (N=1115) | | |
|---|---|---|---|---|---|---|
| | aOR | 95%CI | p-value | aOR | 95%CI | p-value |
| Age ≥65y | 6.64 | (3.51, 12.55) | <0.0001 | 4.22 | (2.51, 7.07) | <0.0001 |
| History of heart failure | | | | 2.29 | (1.19, 4.42) | 0.014 |
| Hypoxia at diagnosis | 2.52 | (1.18, 5.35) | 0.017 | 2.46 | (1.17, 5.17) | 0.017 |
| Non-invasive ventilation at diagnosis | 2.13 | (1.01, 4.53) | 0.049 | 2.67 | (1.28, 5.57) | 0.009 |
| ALC at diagnosis <0.5K/µl | 2.10 | (1.16, 3.79) | 0.014 | 1.98 | (1.14, 3.45) | 0.017 |
| Hemoglobin at diagnosis <10g/dl | 2.40 | (1.30, 4.44) | 0.005 | 1.74 | (0.98, 3.08) | 0.056 |
| Platelet at diagnosis <100K/µl | | | | 2.21 | (1.15, 4.24) | 0.017 |
| LRTI at diagnosis or progression to LRTI | 3.70 | (1.94, 7.08) | <.0001 | 4.16 | (1.95, 8.82) | <0.001 |
| Remdesivir treatment | 0.44 | (0.20, 0.96) | 0.04 | 0.45 | (0.21, 0.98) | 0.04 |

**B) Patients without cancer**

| Independent predictor | Complete Case (CC) (N=1777) | | | Multiple Imputation (MI) (N=2851) | | |
|---|---|---|---|---|---|---|
| | aOR | 95%CI | p-value | aOR | 95%CI | p-value |
| Age ≥65y | 4.91 | (3.39, 7.13) | <.0001 | 4.96 | (3.46, 7.10) | <0.0001 |
| Prior COPD/bronchiolitis obliterans | 1.81 | (1.16, 2.83) | 0.009 | 1.84 | (1.19, 2.84) | 0.006 |
| History of ishemic heart disease | 1.68 | (1.10, 2.56) | 0.017 | 1.69 | (1.12, 2.56) | 0.013 |
| History of hypertension | 1.98 | (1.30, 3.03) | 0.002 | 2.15 | (1.42, 3.24) | <0.001 |
| Hypoxia at diagnosis | 7.53 | (3.72, 15.26) | <0.0001 | 7.91 | (4.22, 14.86) | <0.0001 |
| Mechanical ventilation/intubation | 2.38 | (1.22, 4.62) | 0.011 | 2.15 | (1.13, 4.08) | 0.019 |
| At diagnosis | | | | | | |
| ALC at diagnosis <0.5K/µl | | | | 1.62 | (1.01, 2.59) | 0.044 |
| Creatinine at diagnosis >1.5mg/dL | 1.96 | (1.35, 2.84) | <0.001 | 1.95 | (1.32, 2.87) | <0.001 |
| Coinfection after diagnosis | 3.03 | (1.87, 4.89) | <0.0001 | 2.83 | (1.79, 4.48) | <0.0001 |

ALC=Absolute lymphocyte count; LRTI=Lower respiratory tract infection; COPD=Chronic obstructive pulmonary disease.

The model was adjusted for country and tocilizumb treatment.

The significant differences between the models by CC analysis and by MI analysis were shown in the gray area - (a) History of heart failure and platelet level at diagnosis were independent predictors of 30 day mortality in the multivariable model by MI analysis, but not in the model by CC analysis; (b) Hemoglobin level at diagnosis was an independent predictor of 30 day mortality in the multivariable model by CC analysis, but not in the model by MI analysis.

Hemoglobin level at diagnosis was kept in the final model by MI analysis due to its confounding effect despite its non-significant p-value.

The model was adjusted for country and convalescent plasma treatment.

The significant difference between the models by CC analysis and by MI analysis was shown in the gray area - ALC level at diagnosis was an independent predictor of 30 day mortality in the multivariable model by MI analysis, but not in the model by CC analysis.

rate between patients who received remdesivir and those who did not (29.7 [11 of 37] and 34.4% [53 of 154], respectively; p=0.59).

Since 85% of patients treated with remdesivir also received corticosteroids, we evaluated the impact of their combination therapy on 30 day mortality. Among patients on low-flow oxygen at admission, the mortality rates among those who received remdesivir alone or in combination with corticosteroids (6% [3 of 51]) was significantly lower than the mortality rate of those who received

**Table 4.** 30 day mortality among different groups of COVID-19 patients with cancer.

| Patient group | No. of patients | No. (%) who died within 30 days[†] |
|---|---|---|
| Hematological malignancy | 283 | 37 (13) |
| Transplant within 1y of COVID-19 diagnosis | 15 | 2 (13) |
| Lymphoma or myeloma | 164 | 19 (12) |
| Lymphocytic leukemia (ALL/CLL) | 44 | 6 (14) |
| Myelocytic leukemia | 62 | 8 (13) |
| Solid tumor* | 632 | 47 (7) |
| Lung cancer | 64 | 14 (22) |
| Metastatic non-lung cancer solid tumor | 261 | 17 (7) |
| Non-metastatic non-lung cancer solid tumor | 307 | 16 (5) |

*Patients with missing metastasis data were excluded from the analysis.

[†]30 day mortality comparisons for the groups below. (1) Lung cancer vs metastatic non-lung cancer solid tumor: p< 0.001; (2) Lung cancer vs non-metastatic non-lung cancer solid tumor: p< 0.0001; (3) Hematological malignancy vs lung cancer: p=0.07; (4) Hematological malignancy vs non-lung cancer solid tumor: p< 0.001; (5) None of the above significant differences detected remained significant in multivariable analysis of 30 d mortality in cancer patients.

corticosteroids alone (18.3% [21 of 115]; p=0.036). However, mortality rates were similar for remdesivir alone and combination therapy among various patients' groups (*Supplementary file 1*).

Giving convalescent plasma to patients later than 3d after diagnosis did not make any difference in mortality (data not shown). In contrast to what was observed for convalescent plasma, the later corticosteroids were administered, the greater the benefit. There was a trend toward a greater reduction in 30 day mortality in patients who received corticosteroids later (>5d after diagnosis), whereas no difference was observed in patients who received corticosteroids earlier (*Supplementary file 2*).

## Sensitivity analysis

Lastly, sensitivity analyses were performed to evaluate the impact of missing data on our primary analyses. The multivariable logistic regression models of mortality predictors based on the complete-case analysis and multiple imputation analysis were compared for all patients (*Table 2*) and for patients with and without cancer (*Table 3*). Among all patients, the two models were similar except that multiple imputation analyses did not show remdesivir having a significant impact on mortality while the complete-case analysis did (*Table 2*). However, among patients with cancer, both models showed that remdesivir was independently associated with decreased mortality with similar effects. On the other hand, the two models showed some differences in identifying a few risk factors for 30 day mortality. Among patients without cancer, the two models were similar except that multiple imputation analysis identified one more risk factor – ALC<0.5K/µl at diagnosis (*Table 3*).

## Discussion

Unlike the previously published studies that compared patients with COVID-19 with and without cancer, our study included a large number of patients with cancer from five different continents. Our findings demonstrate that cancer is not an independent risk factor for increased 30 day all-cause mortality in a multivariable logistic regression analyses that accounted for the treatment differences among the countries, which is inconsistent with a recent large study in the US that showed that patients recently receiving cancer treatment had a worse outcome (*Chavez-MacGregor et al., 2022*). The higher mortality rate observed in patients with cancer by univariable analysis seems to be driven by patients with lung cancer and patients with hematological malignancies, a finding that is consistent with prior literature (*Lee et al., 2020*; *Mehta et al., 2020*). Furthermore, lymphopenia which is observed frequently in patients with hematological malignancy was independently associated

with higher COVID-19 mortality according to our country-adjusted multivariable analyses. Two large studies in COVID-19 patients with cancer have shown that lymphopenia was independently associated with increased 30 day mortality (*Lunski et al., 2021*; *Schmidt et al., 2022*). In addition, the mortality rate in the patients with solid tumors other than lung cancer was not different from the mortality rate in the patients without cancer.

Most of the other independent risk factors that we identified for 30 day mortality after a diagnosis of COVID-19 have also been commonly reported as risk factors for mortality in previous studies of COVID-19 in patients with and without cancer with older age (≥65y) being the strongest independent predictor of 30 day mortality in our study (*Zhou et al., 2020*; *Kuderer et al., 2020*; *Rivera et al., 2020*; *He et al., 2020*; *Albiges et al., 2020*; *Robilotti et al., 2020*; *Lee et al., 2020*; *Mehta et al., 2020*; *Tian et al., 2020*; *Lunski et al., 2021*; *Sun et al., 2020*; *Rüthrich et al., 2021*). In our study, the patients with cancer had a more complicated course after discharge from the hospital; specifically, they more frequently required supplemental oxygen and readmission within 30 days after COVID-19 diagnosis.

The presentation pattern of COVID-19 in patients with cancer was different to that of patients without cancer. Patients with cancer seemed to be less symptomatic. This could be related to the fact that patients with cancer tended to be older, with fewer inflammatory cells (neutrophils and lympho-cytes), and more often on corticosteroids. Levels of inflammatory biomarkers were also higher in patients with cancer, particularly D-dimer, ferritin, and procalcitonin.

On multivariable analysis, the only therapeutic agent that independently decreased 30 day all-cause mortality among all patients and in patients with cancer was remdesivir, even after accounting for treatment diffrences among countries. Upon further analysis, remdesivir was found to further decrease mortality in patients with pneumonia and mild hypoxia who were receiving low-flow oxygen (≤6l/min) and not in patients with severe advanced pneumonia who were receiving high-flow oxygen and/or ventilatory support. This is consistent with a large multicenter prospective randomized placebo-controlled trial - Adaptive COVID-19 Treatment Trial-1 (ACTT-1), that found that remdesivir significantly reduced the time to recovery and 28d mortality in patients who were on low-flow oxygen at baseline but not in patients who were on mechanical ventilation or extracorporeal membrane oxygenation at baseline (*Beigel et al., 2020*).

In a meta-analysis that examined four large prospective randomized trials, remdesivir was shown to be associated with reduced 14 d mortality and reduced need for mechanical ventilation (*Shrestha et al., 2021*), which also supports our analysis.

In contrast, the WHO-sponsored multinational Solidarity Trial of COVID-19 hospitalized patients who were randomly assigned to either remdesivir (2750patients) or standard of care (2708patients) showed no difference in overall 28d mortality (*Pan et al., 2021*). However, in two large meta-analyses that each examined more than 13,000 COVID-19 patients from randomized and non-randomized studies, including the WHO randomized trial, remdesivir was associated with a significant improve-ment in the 28d recovery rate (*Rezagholizadeh et al., 2021*; *Lai et al., 2021*). Furthermore, in a large study evaluating treatments and outcomes of COVID-19 among patients with cancer, remdesivir alone was significantly associated with a lower 30 day all-cause mortality rate than other treatments (including high-dose corticosteroids and tocilizumab; *Rivera et al., 2020*). In addition, in a large multi-center matched controlled study involving mostly patients without cancer, Mozaffari et al. demon-strated that remdesivir was significantly effective in reducing mortality in patients on low-flow oxygen but not patients with advanced disease on high-flow oxygen and ventilator support (*Mozaffari et al., 2022*). More recently, a prospective randomized, double blind, placebo-controlled study (involving mainly patients without cancer) showed that early initiation of remdesivir prevents progression to severe COVID-19 (*Gottlieb et al., 2022*).

Therefore, the cumulative data in the literature do support our findings that remdesivir in all patients particularly in patients with cancer improves outcome and may reduce mortality especially if started early in COVID-19 patients with moderate pneumonia who are receiving low-flow oxygen who fit into the stage II of the three-stage COVID-19 classification previously proposed by *Siddiqi and Mehra, 2020*.

The use of corticosteroids (dexamethasone 6mg/d) was shown in a large randomized open-label study conducted in the United Kingdom to be associated with reduced 28d mortality, particularly in patients requiring oxygen supplementation and invasive ventilation who receive corticosteroids after

7d from the onset of symptoms (*Horby et al., 2021*). Subsequently, two meta-analyses of several prospective randomized trials demonstrated that corticosteroid use was significantly associated with a decrease in COVID-19 mortality (*Sterne et al., 2020*; *Siemieniuk et al., 2020*). In our study, we found that if corticosteroids were started more than 5d after the PCR diagnosis of COVID-19, there was a trend toward a reduction in 30 day COVID-19 mortality compared to starting corticosteroids earlier. 5 d after diagnosis by PCR testing would possibly be equivalent to 7d after the onset of symptoms since the average time from symptom onset to PCR diagnosis has been estimated to be 2–3d (*Kucirka et al., 2020*). Furthermore, by multivariable analysis and upon further subanalysis (*Supplementary file 1*), remdesivir was found to improve outcome independently of the effect of steroids.

Our study has several limitations. First, the retrospective design precluded complete assessment of disease progression in the outpatients, which limited their input into the general study. Second, some non-cancer centers contributed data only on their hospitalized patients who were likely sicker. This may have biased the data toward a higher rate of hospitalization among patients without cancer and made the data more heterogeneous. In addition, the predominance of symptomatic patients (87%) might also limit our evaluation of the impact of cancer in the whole picture of the disease. Third, our data contained a lot of missing values due to the nature of data collection. However, we performed sensitivity analyses to evaluate its impact on our primary analyses, and it showed that overall this impact was limited. Most independent predictors were identified by both complete-case and multiple imputation analyses with similar effects. The differences between the two analysis methods in identifying a few independent mortality predictors were probably due to the sample size changes from complete-case analyses to multiple imputation analyses. Last, this study was conducted prior to the introduction of COVID vaccines and included patients who were not vaccinated and who were infected by early variants which limits the generalizability of our results to contemporary COVID-19 patients. A multinational European registry showed that severity of COVID-19 and mortality in cancer patients have improved since 2020. This could be multifactorial related to earlier diagnosis, improved management including current antivirals, monoclonal antibodies, vaccination, as well as different circulating variants of the virus that could be associated with less severe disease than the earlier strains (*Pinato et al., 2022*). However, a recent study comparing vaccinated and unvaccinated patients with cancer showed that despite the protective role of vaccination, vulnerable patients with cancer, particularly those with risk factors such as lymphopenia, active and progressing cancer, and advanced age, can develop severe and fatal breakthrough infections (*Schmidt et al., 2022*).

In conclusion, this is the largest multicenter worldwide study comparing COVID-19 in patients with cancer to those without. In this study, although the limited effect size, underlying malignancy was not found to be an independent risk factor for a higher 30 day all-cause COVID-19 mortality. However, lymphopenia and anemia were frequently observed in patients with hematological malignancies, and patients with lung cancer were associated with the highest risks for poor outcome. Finally, remdesivir stood out as the only therapeutic agent independently associated with decreased 30 day mortality, particularly in patients with cancer on low-flow oxygen. Corticosteroids tended to be most useful if given more than 5d after COVID-19 diagnosis. The role of these therapeutics and their timing of administration should be verified in larger studies, especially in patients with cancer, who tend to have a higher degree of immunosuppression, which may lead to prolongation of the viral phase.

## Acknowledgements

We thank all the investigators, MD Anderson Cancer Network, and the Data-Driven Determinants for COVID-19 Oncology Discovery Effort (D3CODE) Team at The University of Texas MD Anderson Cancer Center for assistance in study development and data extraction. We thank Ms. Meena Medepalli and Mr. Joel Cox from the MD Anderson Cancer Network - External Research at The University of Texas MD Anderson Cancer Center for their support in coordinating the cancer network sites for protocol participation. We thank Mr. Joseph P Thomas at The University of Texas MD Anderson Cancer Center for his assistance with data capture in REDCap, Kris Weaver for data quality review, Sheri Rivera for REDCap build, Regulatory team for the fast activation, Toby and her team for assistance in resolving IT issues with site EMR. We thank Anastasia Turin and Drew Goldstein from Syntropy Technologies LLC, Burlington, MA for interfacing with the Syntropy platform: Palantir Foundry. 'Foundry' is Syntropy's fully-managed, cloud-based software-as-a-service for governing, structuring, and harmonizing real-world data to empower health systems and their collaborators to derive insights from

that data. Editorial assistance was provided by Stephanie P Deming, Research Medical Library at MD Anderson. This assistance was funded by The University of Texas MD Anderson Cancer Center. We thank Ms. Salli Saxton and Ms Christine Cobb at The University of Texas MD Anderson Cancer Center for helping with the submission of the manuscript. This research is supported by the National Institutes of Health/National Cancer Institute under award number P30CA016672, which supports the MD Anderson Cancer Center Clinical Trials Office. National Cancer Institute, National Institutes of Health The funders had no role the design and conduct of the study; collection, management, analysis, and interpretation of the data; preparation, review, or approval of the manuscript; and decision to submit the manuscript for publication.

## Additional information

### Competing interests

Data-Driven Determinants for COVID-19 Oncology Discovery Effort (D3CODE) Team: The other authors declare that no competing interests exist.

### Funding

| Funder | Grant reference number | Author |
| --- | --- | --- |
| National Cancer Institute | | Issam I Raad |
| NIH Clinical Center | | Issam I Raad |
| National Institutes of Health/National Cancer Institute | P30CA016672 | Anne-Marie Chaftari |

The funders had no role in study design, data collection and interpretation, or the decision to submit the work for publication.

### Author contributions

Issam I Raad, Ray Hachem, Conceptualization, Resources, Data curation, Software, Formal analysis, Supervision, Funding acquisition, Validation, Investigation, Visualization, Methodology, Writing – original draft, Project administration, Writing – review and editing; Nigo Masayuki, Conceptualization, Resources, Data curation, Software, Supervision, Validation, Investigation, Visualization, Methodology, Writing – original draft, Project administration, Writing – review and editing; Tarcila Datoguia, Hiba Dagher, Ying Jiang, Alexandre E Malek, Conceptualization, Resources, Data curation, Formal analysis, Supervision, Validation, Investigation, Visualization, Methodology, Project administration, Writing – review and editing; Vivek Subbiah, Conceptualization, Data curation, Formal analysis, Supervision, Validation, Visualization, Writing – review and editing; Bilal Siddiqui, Conceptualization, Supervision, Validation, Visualization, Writing – review and editing; Arnaud Bayle, Conceptualization, Resources, Data curation, Supervision, Validation, Investigation, Visualization, Writing – review and editing; Robert Somer, Conceptualization, Resources, Data curation, Validation, Investigation, Visualization, Methodology, Writing – review and editing; Ana Fernández Cruz, Edward Gorak, Conceptualization, Resources, Data curation, Supervision, Investigation, Methodology, Writing – original draft, Writing – review and editing; Arvinder Bhinder, Conceptualization, Resources, Data curation, Software, Supervision, Validation, Investigation, Visualization, Methodology, Writing – original draft, Writing – review and editing; Nobuyoshi Mori, Nelson Hamerschlak, Conceptualization, Resources, Data curation, Software, Supervision, Validation, Investigation, Visualization; Samuel Shelanski, Conceptualization, Resources, Data curation, Software, Supervision, Validation, Investigation, Visualization, Writing – original draft, Writing – review and editing; Tomislav Dragovich, Suha Fakhreddine, Conceptualization, Resources, Data curation, Supervision, Validation, Investigation, Visualization, Writing – original draft, Writing – review and editing; Yee Elise Vong Kiat, Conceptualization, Resources, Data curation, Validation, Investigation, Visualization, Writing – original draft, Writing – review and editing; Abi Hanna Pierre, Roy F Chemaly, Victor Mulanovich, Javier Adachi, Teny John, Eduardo Yepez Yepez, Ramón Aguado-Noya, Cristina Traseira Puchol, Conceptualization, Resources, Data curation, Supervision, Validation, Investigation, Visualization; Jovan Borjan, Fareed Khawaja, Bruno Granwehr, Cielito C Reyes-Gibby,

Conceptualization, Resources, Data curation, Validation, Investigation, Visualization; Harrys A Torres, Rita E Deeba, Resources, Data curation, Supervision, Validation, Investigation, Visualization; Natraj Reddy Ammakkanavar, Mala Pande, Takahiro Matsuo, Kazuhiro Ishikawa, Ryo Hasegawa, Resources, Data curation, Validation, Investigation, Visualization; Marcel Yibirin, Conceptualization, Resources, Data curation, Validation, Investigation; Noman Ali, Resources, Data curation, Investigation, Visualization; Raniv Dawey Rojo, Data curation, Investigation, Visualization; Shahnoor M Ali, Data curation, Validation, Investigation, Visualization; Patrick Chaftari, Resources, Data curation, Supervision, Investigation, Visualization; Alvaro Garcia García, Dong Gun Lee, Conceptualization, Resources, Data curation, Software, Supervision, Validation, Investigation, Visualization, Methodology; Monica Slavin, Conceptualization, Resources, Data curation, Supervision, Validation, Investigation, Visualization, Methodology; Benjamin Teh, Conceptualization, Resources, Data curation, Formal analysis, Supervision, Validation, Investigation, Visualization, Methodology; Cesar A Arias, Conceptualization; Dimitrios P Kontoyiannis, Resources, Data curation, Investigation; Anne-Marie Chaftari, Conceptualization, Data curation, Software, Formal analysis, Supervision, Funding acquisition, Validation, Investigation, Visualization, Methodology, Writing – original draft, Project administration, Writing – review and editing

### Author ORCIDs
Hiba Dagher ⬤ http://orcid.org/0000-0002-8483-6977
Nobuyoshi Mori ⬤ http://orcid.org/0000-0001-8815-5135
Yee Elise Vong Kiat ⬤ http://orcid.org/0000-0002-8870-1391
Cielito C Reyes-Gibby ⬤ http://orcid.org/0000-0003-4500-6476
Anne-Marie Chaftari ⬤ http://orcid.org/0000-0001-8097-8452

### Ethics
Human subjects: This study (Protocol # 2020-0437) was approved by the institutional review board at MD Anderson Cancer Center and the institutional review boards of the collaborating centers. A patient waiver of informed consent was obtained.

### Decision letter and Author response
Decision letter https://doi.org/10.7554/eLife.81127.sa1
Author response https://doi.org/10.7554/eLife.81127.sa2

## Additional files

### Supplementary files
• Supplementary file 1. Comparing mortality in patients treated with remdesivir alone or with steroids for COVID-19.

• Supplementary file 2. Timing of administration of corticosteroids as COVID-19 treatment and 30 d mortality.

• MDAR checklist

### Data availability
We are unable to share the data given our restriction policy and the fact that this study includes data from 16 centers and from the five continents. We do not have an agreement or the permission to share our data and other centers' data. All the analyses were performed using SAS version 9.4 (SAS Institute Inc, Cary, NC).

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
