## [Editor Report]

This study addresses the important subject of correlates of COVID-19 mortality in patients with cancer. It is a large multinational cohort of patients. The authors provide solid evidence based on careful multivariate analysis for the finding that cancer is not an independent risk factor for COVID-19 death.

---

## [Decision Letter]

**Decision letter after peer review:**

Thank you for submitting your article "International Multicenter Study Comparing Cancer to Non-Cancer Patients with COVID-19: Impact of Risk Factors and Treatment Modalities on Survivorship" for consideration by *eLife*. Your article has been reviewed by 2 peer reviewers, and the evaluation has been overseen by a Reviewing Editor and Eduardo Franco as the Senior Editor. The following individual involved in the review of your submission has agreed to reveal their identity: Annika Fendler (Reviewer #1).

As is customary in *eLife*, the reviewers have discussed their critiques with one another and with the editors. What follows below is our edited compilation of the essential and ancillary points provided by reviewers in their critiques and in their interaction post-review. Please submit a revised version that addresses these concerns directly. Although we expect that you will address these comments in your response letter, we also need to see the corresponding revision clearly marked in the text of the manuscript. Some of the reviewers' comments may seem to be simple queries or challenges that do not prompt revisions to the text. Please keep in mind, however, that readers may have the same perspective as the reviewers. Therefore, it is essential that you attempt to amend or expand the text to clarify the narrative accordingly.

Essential revisions:

(1) Consort diagram to explain attrition.

(2) Sensitivity analysis to evaluate the impact of data missingness.

(3) Justify clearly the choice of multivariate models.

4) Detail the cause of death as per reviewer 1.

(5) Discuss the fact this is an unvaccinated cohort and how that has implications in the context of vaccination.

*Reviewer #1 (Recommendations for the authors):*

– The data contain a lot of missing values, likely due to the nature of data collection. The authors should perform a sensitivity analysis to explore how this affects their results.

– The authors mention that patients were lost to follow-up especially those not hospitalised. I'm not clear how many patients are in both groups. I also can't see how many patients were censored in the Kaplan-Meyer Analysis. A consort diagram would be helpful to understand how many patients the primary endpoint could be investigated.

– The authors present several multivariate models. It is not clear how they chose which data to include and on what basis the variables were selected.

– The authors describe that 348 patients died, and 93% of death are COVID-related. How was this split between patients with cancer and those without? Was there a difference in the proportion of death that was COVID-19-related?

– An overview of all treatments that patients received is missing. Were these similar between patients with cancer and those without? Did they differ by country? Did this impact outcome?

– The discussion is very long and includes a detailed description of published studies. The authors should summarise the findings of previously published studies in a concise way and restrict this discussion only to the direct results presented in the paper. For example, the authors discuss in length which treatment should be given at a certain stage of COVID-19, which is not directly relevant to the data they present.

– The authors only briefly discuss that this study was conducted in an unvaccinated cohort. They should include a more detailed discussion of and how their data apply to the current situation.

*Reviewer #2 (Recommendations for the authors):*

Suggestions to the authors before considering this paper for publication:

(1) Replace the term "cancer patients and non-cancer patients" with "patients with and without cancer".

(2) It is unclear what tables 2 and 3 add to the manuscript and the authors might want to consider moving them to the supplement.

(3) The table of patients at risk shown in Figure 1 should be adjusted to include data at some, but not all time points.

4) The authors may want to cite PMID: 34958894 to discuss the impact of cancer on patients with COVID-19 in the era of COVID-19 vaccines.

---

## [Author Response]

Essential revisions:1) Consort diagram to explain attrition.

We have added a consort diagram to explain patient attrition as shown in figure 1.

2) Sensitivity analysis to evaluate the impact of data missingness.

We performed a sensitivity analysis to evaluate the impact of data missingness on our primary data analysis. Please see below the detailed response to the reviewer.

3) Justify clearly the choice of multivariate models.

We have provided a detailed description of the performance of our multivariable logistic regression analyses including variable selection, univariable analysis, multivariable analysis and the final models. Please see below the detailed response to the reviewer’s comment.

(4) Detail the cause of death as per reviewer 1.

We have revised the manuscript to reflect that any death that occurred within 30 days after COVID-19 diagnosis was considered to be COVID-related, irrespective of other comorbidities that could have contributed to death.

(5) Discuss the fact this is an unvaccinated cohort and how that has implications in the context of vaccination.

We agree with the reviewer and have acknowledged this limitation in the Discussion section as follows: “Last, this study was conducted prior to the introduction of COVID vaccines and included patients who were not vaccinated and who were infected by early variants which limits the generalizability of our results to contemporary COVID-19 patients”. We have added in the Discussion section the results of a recent study on the impact of vaccination in cancer patients.

Reviewer #1 (Recommendations for the authors):– The data contain a lot of missing values, likely due to the nature of data collection. The authors should perform a sensitivity analysis to explore how this affects their results.

We totally agree with the reviewer. This study aimed to identify risk factors for 30-day mortality of COVID-19 patients and the data had a lot of missing values. Following this advice, we performed a sensitivity analysis to evaluate the impact of missing data on our primary data analysis. We first estimated missing values using multiple imputation technique and then performed a multivariable analysis based on the imputed datasets. Lastly, the analysis results were compared between our initial primary analysis based on the complete cases (patients with no missing values for the variables retained for the final multivariable model) and the analysis based on the imputed datasets. We have added the sensitivity analyses results in the revised manuscript in Table 3 (for all patients) and Table 4 (for patients with (A) and without cancer (B)) and summarized the main findings in Results section. We also included a description of the sensitivity analysis in the revised Statistical analysis section.

– The authors mention that patients were lost to follow-up especially those not hospitalised. I'm not clear how many patients are in both groups. I also can't see how many patients were censored in the Kaplan-Meyer Analysis. A consort diagram would be helpful to understand how many patients the primary endpoint could be investigated.

We are sorry, we did not make a statement about loss to follow-up in the manuscript and there might be a misunderstanding about it. We mentioned “some non-cancer centers contributed data only on their hospitalized patients who were likely sicker” because some non-cancer centers provided only hospitalized (non-cancer) patients to this study. We considered it a study limitation in Discussion section because of a possible imbalance in COVID-19 infection severity between patients with and without cancer in our study population. Regarding loss to follow up, although our data had a lot of missing values for many variables, the data for the primary endpoint 30-day mortality were relatively complete, and there were only 10 patients excluded from the Kaplan-Meier analysis due to their missing death dates, including 8 patients with cancer and 2 patients without cancer. However, after redoing multivariable analysis adjusting for country as advised by another comment, we found that cancer was no longer independently associated with 30-day mortality. Therefore, the Kaplan-Meier showing a significant impact of cancer on mortality has been no longer appropriate for this manuscript and therefore, we have removed this figure form the revised manuscript. As advised, a consort diagram of patient attrition has been added to the manuscript (Please see Figure 1).

– The authors present several multivariate models. It is not clear how they chose which data to include and on what basis the variables were selected.

We thank the reviewer for this comment and have modified the Statistical Analysis section by providing a detailed description of the performance of our multivariable logistic regression analyses including variable selection, univariable analysis, multivariable analysis and the final models.

– The authors describe that 348 patients died, and 93% of death are COVID-related. How was this split between patients with cancer and those without? Was there a difference in the proportion of death that was COVID-19-related?

We have revised the manuscript to reflect that any death that occurred within 30 days after COVID-19 diagnosis was considered to be COVID-19-related, irrespective of other comorbidities that could have contributed to death. We have revised the manuscript accordingly.

– An overview of all treatments that patients received is missing. Were these similar between patients with cancer and those without? Did they differ by country? Did this impact outcome?

Treatment data are an important part of this study and we apologize for missing to provide them. We have added the treatment data in Table 1. As we can see, there was a significant difference in every treatment received between patients with and without cancer. Treatments that patients received also significantly differed by country and we have added this finding in Results section. In addition, to account for the treatment differences among the countries, we redid multivariable logistic regression analyses on mortality with adjusting for the country effect. Compared to our previous analysis, a main change in the new analyses is that cancer was no longer an independent risk factor for 30-day mortality after adjusting for country in multivariable analysis. We believe that lymphopenia and possibly anemia which occurred more frequently in cancer patients (Table 1) might be the confounding independent variables that made cancer appear to be associated with higher mortality by univariate analysis. Another change is that among patients without cancer, remdesivir was no longer independently associated with 30-day mortality, while it still had a similar significant impact on reducing mortality among all patients and among patients with cancer. We have updated the analyses results in the revised manuscript.

– The discussion is very long and includes a detailed description of published studies. The authors should summarise the findings of previously published studies in a concise way and restrict this discussion only to the direct results presented in the paper. For example, the authors discuss in length which treatment should be given at a certain stage of COVID-19, which is not directly relevant to the data they present.– The authors only briefly discuss that this study was conducted in an unvaccinated cohort. They should include a more detailed discussion of and how their data apply to the current situation.

We have shortened our discussion to mainly focus on our results. We agree with the reviewer that this study was conducted in an unvaccinated cohort before the introduction of COVID vaccines and have acknowledged this limitation in the Discussion section as follows: “Last, this study was conducted prior to the introduction of COVID vaccines and included patients who were not vaccinated and who were infected by early variants which limits the generalizability of our results to contemporary COVID-19 patients”. We have added in the Discussion section the results of a recent study on the impact of vaccination in cancer patients.

Reviewer #2 (Recommendations for the authors):Suggestions to the authors before considering this paper for publication:(1) Replace the term "cancer patients and non-cancer patients" with "patients with and without cancer".

We have replaced the term “cancer patients and non-cancer patients” with “patients with and without cancer” as requested throughout the manuscript.

(2) It is unclear what tables 2 and 3 add to the manuscript and the authors might want to consider moving them to the supplement.

We thank the reviewer for their comments. We have shortened table 2 and moved table 3 to supplement Table 1. We have added additional tables and a new figure to address the other comments.

(3) The table of patients at risk shown in Figure 1 should be adjusted to include data at some, but not all time points.

We agree and thank the reviewer for the advice. However, after redoing multivariable analysis adjusting for country as advised in another comment, we found that cancer was no longer independently associated with 30-day mortality. Therefore, the Kaplan-Meier showing a significant impact of cancer on mortality has been no longer appropriate for this manuscript and therefore, we have removed this figure form the revised manuscript.

(4) The authors may want to cite PMID: 34958894 to discuss the impact of cancer on patients with COVID-19 in the era of COVID-19 vaccines.

We have cited this reference and added the findings of the impact of vaccination in patients with cancer in the Discussion section.